# LC-QAT: Data-Efficient 2-Bit QAT for LLMs via Linear-Constrained Vector Quantization

**Haoyu Wang** [* 1] **Xingyu Yu** [* 2] **Haiyan Zhao** [† 1] **Fengxiang Wang** [3] **Xu Han** [† 1]

## Abstract

Quantization-aware training (QAT) is essential for extremely low-bit large language models (LLMs). Current QAT methods are mainly based on scalar quantization (SQ), which enables efficient optimization but suffers from severe performance degradation at 2-bit precision. On the other hand, vector quantization (VQ) provides substantially higher representational capacity, but its discrete codebook lookup prevents end-to-end training. We propose LC-QAT, a 2-bit weight-only VQ-QAT framework that represents quantized weights via a learned affine mapping over discrete vectors, which yields a high-quality PTQ initialization and enables fully differentiable end-to-end optimization **without explicit codebook lookup in the training forward pass**. This strong post-training initialization makes LC-QAT highly data-efficient. Experiments across diverse LLMs demonstrate that LC-QAT consistently outperforms state-of-the-art QAT methods while using only **0.1%–10% of the training data**. Our results establish LC-QAT as a practical and scalable solution for extreme low-bit model deployment.

## 1. Introduction

Large language models (LLMs) have achieved remarkable success across various tasks, yet their substantial memory and computational requirements pose challenges for deployment on resource-constrained devices. Model quantization (Frantar et al., 2022; Egiazarian et al., 2024) has therefore become a key technique for enabling efficient inference, es-pecially in extremely low-bit regimes such as 1–2 bits (Hao et al., 2025; Chee et al., 2023; Baalen et al., 2024; Zhou et al., 2025; Tseng et al., 2024b).

Existing quantization methods are commonly divided into Post-Training Quantization (PTQ) (Frantar et al., 2022; Egiazarian et al., 2024) and Quantization-Aware Training (QAT) (Ma et al., 2025; Liu et al., 2023). In the aggressive 2-bit setting, QAT consistently outperforms PTQ by adapting model parameters to compensate for quantization errors (Liu et al., 2025). However, most existing QAT frameworks rely on scalar quantization (Ma et al., 2025; Egiazarian et al., 2024). These SQ-based QAT works usually independently quantize and dequantize each weight to include quantization error during training. While being easy to optimize, SQ-based QAT suffers from severe information loss at ultra-low precision, leading to weak initializations and a heavy dependence on massive training data for recovery.

In contrast, Vector Quantization represents groups of weights using entries from a shared codebook and offers substantially stronger representational capacity under 2-bit constraints. By assigning each group of weights to one of the possible codewords, VQ-based methods preserve significantly more information than SQ and achieve much higher post-quantization accuracy (Egiazarian et al., 2024; Baalen et al., 2024; Zhou et al., 2025). However, incorporating VQ into end-to-end QAT remains highly challenging. The core difficulty lies in the time-consuming nearest-neighbor search during quantization and discrete codebook lookup during dequantization. Existing attempts to address this issue rely on expensive coordinate descent or beam search procedures (Malinovskii et al., 2024), resulting in inefficient and unsynchronized updates of model parameters.

In this work, we propose LC-QAT, a novel end-to-end vector quantization-aware training framework for 2-bit weight-only quantization that overcomes this fundamental limitation. Our key idea is to replace unconstrained codebooks with a linear-constrained parameterization. As shown in Figure 1, each codeword in a weight matrix is generated by applying a shared linear mapping to a discrete quaternary vector. This reformulation transforms discrete index selection into simple rounding and clamping operations followed by linear projection, enabling gradients to propagate through the

---

[*]Equal contribution † Corresponding author [1]Department of Computer Science and Technology, Tsinghua University, Beijing, China [2]School of Software and Microelectronics, Peking University, Beijing, China [3]College of Computer Science and Technology, National University of Defense Technology, Hunan, China. Correspondence to: Xu Han <han-xu@tsinghua.edu.cn>, Haiyan Zhao <zhao_haiyan@foxmail.com>.

*Proceedings of the 43 rd International Conference on Machine Learning*, Seoul, South Korea. PMLR 306, 2026. Copyright 2026 by the author(s).

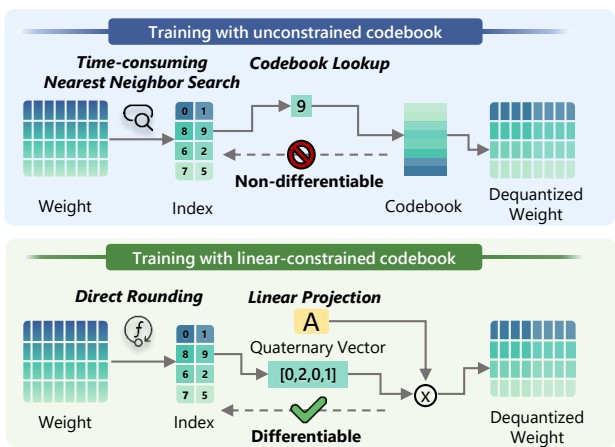

*Figure 1.* LC-QAT training pipeline with a linear-constrained parameterization. By replacing discrete codebook lookup with an SQ-style round/clip discretization followed by an affine projection, LC-QAT makes VQ-QAT lookup-free in the forward pass and compatible with standard end-to-end backpropagation.

quantization process without explicit index search. As a result, LC-QAT makes vector-quantized weights trainable under standard backpropagation.

Building upon a high-quality 2-bit initialization, LC-QAT enables efficient fine-tuning while significantly reducing the data requirements. Experiments on multiple large-scale LLMs and benchmarks demonstrate that our method matches or surpasses state-of-the-art SQ- and VQ-based QAT approaches using only 0.1%–10% of training data.

In summary, our main contributions are as follows:

- We propose a lookup-free parameterization for 2-bit VQ-QAT that removes explicit codebook lookup in training and enables end-to-end optimization with standard backpropagation.

- We provide empirical evidence that LC-QAT starts in a substantially more favorable optimization region than SQ-based 2-bit QAT, which helps explain its improved trainability and data efficiency.

- Extensive experiments show that LC-QAT achieves higher final accuracy than prior VQ-QAT methods and substantially better data efficiency than SQ-QAT methods, with consistent gains as the training budget increases across diverse benchmarks.

## 2. Preliminaries and Initialization

In this section, we first introduce the formulation of vector quantization for neural network weights, and then describe the proposed linear-constrained codebook and the initializa-

tion procedure adopted in LC-QAT. Finally, we present a preliminary analysis of the resulting optimization landscape.

### 2.1. Vector Quantization for Neural Network Weights

Consider a weight matrix $W \in \mathbb{R}^{m \times n}$ in a linear layer. In vector quantization, the matrix is partitioned into groups of size $d$ along predefined dimension $n$. Let $w_g \in \mathbb{R}^d$ denote the $g$-th weight group. A shared codebook

$$C = \{c_1, c_2, \ldots, c_K\}, \quad c_k \in \mathbb{R}^d,$$

is used to represent all groups, where $K$ denotes the number of codewords.

For each group $w_g$, VQ assigns an index by solving the nearest-neighbor problem

$$i_g = \arg \min_{k \in \{1, \ldots, K\}} \|w_g - c_k\|_2^2, \tag{1}$$

and the quantized weight is given by

$$\hat{w}_g = c_{i_g}. \tag{2}$$

In the 2-bit setting considered in this work, each dimension takes 4 discrete values, resulting in $K = 4^d$ possible codewords. Compared with scalar quantization, which restricts each weight to only 4 values, VQ provides substantially higher representational capacity.

Despite its strong expressive power, the discrete assignment in Equation (1) is inherently non-differentiable with respect to the index $i_g$. As a result, standard gradient-based optimization cannot directly update the codebook indices during training, posing a fundamental obstacle to end-to-end quantization-aware training.

### 2.2. Initialization with the Linear-Constrained Codebook

To overcome the non-differentiability of conventional codebook lookup, we extend the PTQ method in (Wang et al., 2026) to QAT and parameterize codewords using a linear transformation that is shared within a weight matrix.

Specifically, each codeword is generated as

$$c = Az + B, \tag{3}$$

where $z \in \{0, 1, 2, 3\}^d$ is a discrete quaternary vector, and $A \in \mathbb{R}^{d \times d}$ and $B \in \mathbb{R}^d$ are floating-point parameters.

This formulation defines a structured codebook in which all codewords lie in an affine subspace determined by $A$ and $B$. By enumerating all possible values of $z$, the resulting codebook implicitly contains $4^d$ entries.

Compared with unconstrained codebooks, the proposed linear-constrained parameterization eliminates the need for

nearest-neighbor search. Instead, the discrete vector $z$ can be obtained through rounding and clamping operations, and gradients can propagate through the linear mapping in Equation (3). The main difference between the proposed linear-constrained method and lattice-based methods, such as Quip# (Tseng et al., 2024a) and NestQuant (Savkin et al., 2025), is that the former can generate all codewords from the same linear transformation. However, the padding codewords in Quip# and the nested codebooks in NestQuant cannot meet this condition. While the latter offers more flexibility, it necessitates codebook lookup.

In Equation (3), $A$ and $B$ are initialized as follows:

$$A = sG, \quad B = -sG\mu\mathbf{1}, \quad (4)$$

where $\mathbf{1}$ denotes an all-ones vector, $G \in \mathbb{R}^{d \times d}$ is a random orthogonal matrix, $\mu = (2^b - 1)/2$ is a centering constant, and $s = \sqrt{12/(2^{2b} - 1)}$ is a scaling factor. This initialization ensures that the codebook has approximately zero mean, unit variance, and weak inter-dimensional correlation.

We adopt the LDLQ algorithm (Chee et al., 2023) to generate the initial quantized model and the Hadamard transformation for eliminating the outliers following QuIP# (Tseng et al., 2024a), QTIP (Tseng et al., 2024b), YAQA (Tseng et al., 2025) and NestQuant (Savkin et al., 2025). LDLQ performs column-wise group quantization and sequentially compensates for reconstruction errors.

### 2.3. Preliminary Optimization Analysis

We empirically analyze the quality of the proposed initialization by the loss landscape around the initial points.

Following Chen et al. (2025), we project the loss surface of the Qwen-3-1.7B model onto two given directions and evaluate the cross-entropy loss on WikiText-2 (Merity et al., 2016). Figure 2a illustrates the resulting landscapes for LC-QAT and SQ-based QAT. It is shown that the LC-QAT initialization is close to the FP16 baseline, indicating significantly lower performance degradation. In contrast, the SQ-based initialization deviates substantially from the global minimum. We further validate the zero-shot QA accuracy of VQ and SQ initial models in Appendix A.3.

Figure 2b shows that the LC-QAT initialization lies in the low-loss basin and exhibits a saddle-point structure similar to that of the full-precision model. In contrast, Figure 2c shows that SQ-based initialization deviates substantially from the optimal region and lacks a nearby local minimum.

This phenomenon can be attributed to the fact that vector quantization preserves more information during post-training compression. As a result, LC-QAT begins optimization from a more favorable region of the parameter space, leading to reduced optimization difficulty and improved data efficiency in subsequent fine-tuning.

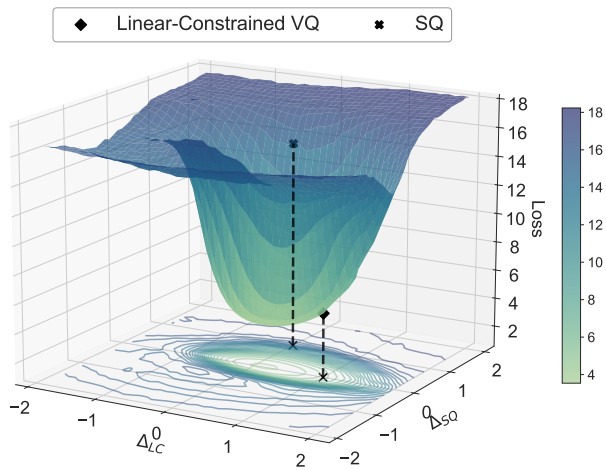

*(a)* Loss landscape of the FP16 Model, with the initial loss of LC-QAT and a SQ-Based QAT model.

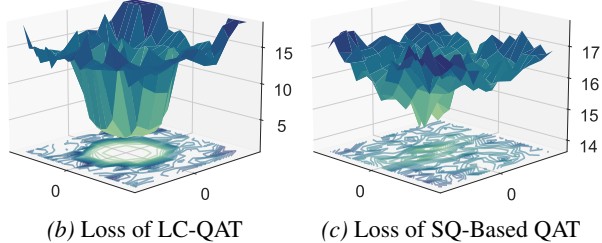

*(b)* Loss of LC-QAT  *(c)* Loss of SQ-Based QAT

*Figure 2.* (a) The loss landscape of a FP16 Qwen-3-1.7B model and the initial point of the LC-QAT and SQ-based QAT models. The loss is measured by cross entropy on the WikiText-2 dataset. LC-QAT closely approaches the local minimum, whereas the SQ model remains distant from the optimal region. (b) The loss landscape for the LC-QAT model. The surface exhibits a distinct saddle point structure, with the training starting point $(0,0)$ positioned significantly closer to a local minimum. (c) The loss landscape for the SQ-based QAT model. The surface has higher overall loss values and the absence of a well-defined local minimum, indicating a more challenging optimization landscape compared to LC-QAT.

## 3. Method

In this section, we present the proposed LC-QAT framework for end-to-end training of VQ LLMs. We first provide an overview of the training pipeline. We then introduce the forward pass and the differentiable gradient estimator. Finally, we describe the improvement of training stability.

### 3.1. Overview of LC-QAT

As shown in Figure 3, LC-QAT represents quantized weights using discrete integer variables and a linear transformation. During training, these integer variables are obtained by the discretization of continuous proxy weights. Gradients are propagated through the discretization using differentiable approximations.

Specifically, LC-QAT maintains a set of floating-point proxy weights $W_p \in \mathbb{R}^{m \times n}$. In the forward pass, $W_p$ is converted into integer weights $W_z$ via rounding and clamping.

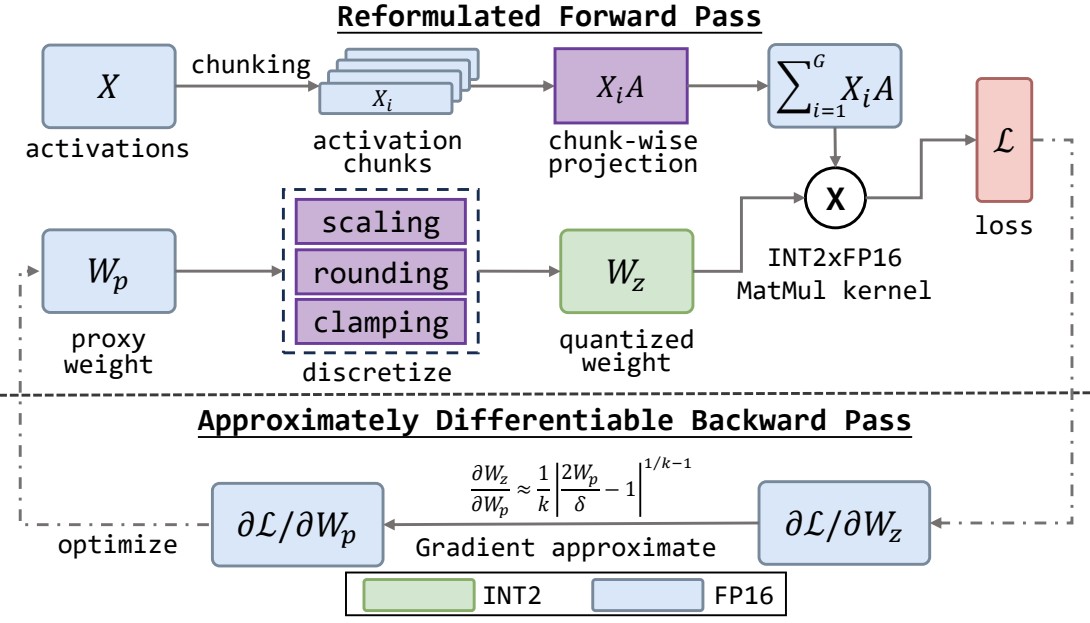

*Figure 3.* Overview of the forward and backward pass of LC-QAT. During the forward pass, proxy weights are discretized into integer weights to incorporate quantization errors. The computational workflow is reformulated to leverage Int2-FP16 MatMul kernels, which are well-optimized for SQ models. In the backward pass, by bypassing the traditional codebook lookup operation, LC-QAT enables end-to-end optimization via approximate gradients. LC-QAT utilizes a Differentiable Gradient Estimator (DGE) to facilitate stable gradient flow for the integer weights.

The integer weights are then decoded into effective weights $\hat{W}$ through a linear-constrained codebook. The resulting weights are used to compute outputs and training loss.

During backpropagation, approximate gradient estimators are employed to propagate gradients through the discretization operation, enabling end-to-end optimization of $W_p$.

### 3.2. The forward pass of LC-QAT

Given proxy weights $W_p \in \mathbb{R}^{m \times n}$, the corresponding integer weights are computed as

$$W_z = \text{clip}\left(\text{round}(W_p),\, 0,\, 2^b - 1\right), \qquad (5)$$

where $b$ denotes the quantization bit-width, and $\text{clip}(\cdot)$ denotes element-wise clamping.

The above process is basically the same as that of SQ-based QAT. However, LC-QAT does not treat the elements in $W_z$ as integer approximations of floating-point weights, but rather regards them as codewords in the linear-constrained codebook and performs differentiable dequantization through a linear mapping. We partition $W_z$ into $G = n/d$ groups along the column dimension:

$$W_z = [W_{z,1}, W_{z,2}, \ldots, W_{z,G}], \qquad (6)$$

where each block $W_{z,i} \in \mathbb{Z}^{m \times d}$ contains $d$ consecutive

columns. As a result, the decoding procedure can be conducted as follows:

$$\hat{W}^T = \begin{bmatrix} A W_{z,1}^T + B\mathbf{1}^T \\ A W_{z,2}^T + B\mathbf{1}^T \\ \vdots \\ A W_{z,G}^T + B\mathbf{1}^T \end{bmatrix} \qquad (7)$$

Equation (7) eliminates explicit nearest-neighbor search and enables direct gradient propagation through the linear mapping. Moreover, it can be re-formulated to reduce computational overhead. Given input activations $X \in \mathbb{R}^{N \times n}$, the output $Y = X\hat{W}$ is computed as:

$$
\begin{aligned}
Y &= \sum_{i=1}^{G} X_i (A W_{z,i}^T + B\mathbf{1}^T) \\
&= \sum_{i=1}^{G} (X_i A) W_{z,i}^T + \sum_{i=1}^{G} (X_i B)\mathbf{1}^T
\end{aligned} \qquad (8)
$$

In Equation (8), $X_i A$ corresponds to a floating-point activation, whereas $W_{z,i}$ remains an integer matrix. This formulation allows LC-QAT to reuse optimized integer matrix multiplication kernels designed for scalar quantization.

While the introduction of linear mapping in LC-QAT incurs

additional computation, this overhead remains controllable. For $X \in \mathbb{R}^{N \times n}$ and $W_z \in \mathbb{Z}^{m \times n}$, the dequantization incurs a computational cost of $O(Nnd) + O(Nn)$. Given that the group size $d$ is typically small (e.g., 4 or 8), this additional overhead is minimal relative to the standard $O(Nnm)$ matrix multiplication. Using our custom CUDA kernel, we achieved a 1.68x speedup compared to the FP16 baseline and a 1.43x speedup compared to the AQLM quantized model. Detailed throughput measurements are provided in Appendix A.2.

### 3.3. Differentiable Gradient Estimation

The discretization operation in Equation (5) is non-differentiable, as the derivative of the rounding function is zero almost everywhere. A common approach is to employ the Straight-Through Estimator (STE), which approximates $\partial W_z / \partial W_p \approx 1$.

However, this estimation is inherently imprecise. The resulting gradient noise can introduce significant instability near local minima (Yin et al., 2019), posing a severe threat to the convergence of LLMs (Wang et al., 2025). While prior SQ-based QAT methods have successfully employed STE (Ma et al., 2025; Liu et al., 2025), we hypothesize this is because the substantial parameter modification of 2-bit SQ can effectively push the model away from its initial local minima. In contrast, LC-QAT begins with a better initialization and may be more strongly affected by the instability of STE near the local minimum. To address this issue, we adopt the Differentiable Gradient Estimator (DGE) proposed in (Wang et al., 2025). Specifically, the discretization function is approximated by

$$f(x) = \frac{\delta}{2} \left( 1 + \text{sign}\left( \frac{2x}{\delta} - 1 \right) \left| \frac{2x}{\delta} - 1 \right|^{1/k} \right), \quad (9)$$

where $\delta$ denotes the quantization interval and $k$ controls the sharpness of the approximation.

The derivative is given by

$$f'(x) = \frac{1}{k} \cdot \left| \frac{2x}{\delta} - 1 \right|^{1/k - 1} \quad (10)$$

As $k$ increases, $f(x)$ approaches the hard clamp function, while maintaining smooth gradients for backpropagation.

### 3.4. Integer Weight Preprocessing

In standard SQ, the floating-point weights $W_p$ in Equation (5) are typically initialized randomly (Ma et al., 2025) or from a pre-trained model (Liu et al., 2023; Chen et al., 2024; Liu et al., 2025), with $W_z$ naturally being the quantization of $W_p$. However, this does not hold for LC-QAT. Instead, $W_z$ are derived from a preceding PTQ process.

Consequently, we must reverse this logic: $W_z$ determines the initialization of $W_p$. As a result, we use the following affine transformation for initialization:

$$W_p = s(W_z - t), \quad (11)$$

where

$$s = \frac{2a}{2^b - 1}, \quad t = \frac{2^b - 1}{2}, \quad a = \sqrt{\frac{6}{m+n}}. \quad (12)$$

Equation (11) aligns the statistics of $W_p$ with Xavier initialization, ensuring approximately zero mean and layer-dependent variance. This design preserves inference consistency and improves training stability.

## 4. Experimental Setup

### 4.1. Experimental Protocols

We evaluate LC-QAT under two experimental protocols designed to assess its model capacity and data efficiency.

**(1) Model Capacity Experiments.** We compare LC-QAT with existing VQ-QAT methods using the same amount of data. These experiments focus on assessing LC-QAT's ability to improve performance by scaling the dataset through end-to-end training and synchronous updates of all VQ-quantized parameters.

**(2) Data Efficiency Experiments.** We compare LC-QAT with state-of-the-art scalar-quantized QAT methods. Since most of these methods focus on the foundation models, namely the base models, we set a group of experiments to evaluate the performance degradation of LC-QAT on these foundation models. We then evaluate LC-QAT in the instruction-tuning setting and compare it with instruction-following low-bit models. These experiments aim to evaluate the performance of quantized LLMs in real-world applications and assess whether LC-QAT preserves instruction-following and reasoning abilities under 2-bit quantization.

### 4.2. Models and Baselines

We select Qwen-3-1.7B and 8B (Yang et al., 2025) along with LLaMA-3-3B and 8B (Llama Team, 2024) as our primary foundation models.

For the model capacity experiments, we compare LC-QAT with PV-Tuning (Malinovskii et al., 2024), which, to our knowledge, is the only end-to-end QAT framework for VQ.

For the data efficiency experiments, we also include SQ-based QAT foundation models such as LLM-QAT (Liu et al., 2023), EfficientQAT (Chen et al., 2024) and ParetoQ (Liu et al., 2025), which is the state-of-the-art 2-bit QAT to our knowledge. Since the results of ParetoQ are reported on LLaMA-3, we conduct a controlled experiment on

the same model family to ensure a fair comparison. For the instruction-tuned QAT models, we compare our model with BitNet 2B4T (Ma et al., 2025), which is a high-performance instruction-following QAT model. Unlike LC-QAT, BitNet involves training from scratch using a massive 4T-token corpus to achieve low-bit quantization.

### 4.3. Datasets and Evaluation Benchmarks

**Training Data.** For VQ-QAT and standard QAT on foundation models, we randomly sample approximately 4 billion tokens from the FineWeb dataset (Penedo et al., 2024). Instruction-tuned QAT experiments are conducted on the AM-Qwen3-Distilled dataset with approximately 2B tokens. Example training samples are provided in Appendix A.1.

**Calibration Data.** For post-training VQ, Hessian statistics required by LDLQ are computed using 1,024 samples from the RedPajama dataset (Weber et al., 2024).

**Evaluation Benchmarks.** We categorize our evaluation into 2 groups to align with the baselines:

- **Zero-Shot QA:** These evaluations are used to evaluate the VQ-QAT models and foundation models, including ARC-Easy, ARC-Challenge (Clark et al., 2018), BoolQ (Clark et al., 2019), HellaSwag (Zellers et al., 2019), PIQA (Bisk et al., 2020), and WinoGrande (Sakaguchi et al., 2021). The setting is similar to the zero-shot evaluation of ParetoQ (Liu et al., 2025) and PV-Tuning (Malinovskii et al., 2024).

- **Instruction-Following Tasks:** These evaluations are used to assess knowledge capabilities, instruction-following ability, as well as mathematical and coding problem-solving skills. We select OpenBookQA (Mihaylov et al., 2018), IFEval (Zhou et al., 2023), MMLU (Hendrycks et al., 2021), GSM8K (Cobbe et al., 2021), MATH (Lightman et al., 2023), and HumanEval (Chen et al., 2021) for evaluation.

### 4.4. Implementation Details

**Quantization Configuration.** We set the vector dimension to $d = 4$, resulting in $A \in \mathbb{R}^{4 \times 4}$ and $B \in \mathbb{R}^4$. This configuration introduces only 20 additional parameters per weight matrix. Quantization is applied to all the linear layers in all the transformer blocks.

**Training Setup.** All experiments are conducted on 16 NVIDIA A100 GPUs. For foundation models and VQ-QAT, we use a batch size of 8 with gradient accumulation of 8, resulting in an effective batch size of 1024 sequences. Models are trained for approximately 1,900 steps.

The learning rate is set to $1 \times 10^{-4}$ for Qwen-3-1.7B and LLaMA-3-3B, and $3 \times 10^{-5}$ for Qwen-3-8B and LLaMA-

3-8B. For instruction-tuned QAT, we use a learning rate of $1 \times 10^{-5}$ and gradient accumulation of 8.

**Optimization.** We use AdamW with standard hyperparameters and the warmup-stable-decay learning rate scheduler (Hu et al., 2024). All experiments employ gradient clipping with a threshold of 1.0. We set $\delta = 1$ and $k = 5$ in Equation (9).

## 5. Results

This section mainly analyzes the experiments presented in the previous chapter to demonstrate that our approach offers superior learning capability compared to conventional VQ-QAT methods and a significant data efficiency advantage over state-of-the-art SQ-QAT methods. Additionally, we include two ablation studies to validate the effectiveness of our preprocessing method used before training and the gradient approximation technique employed during training.

### 5.1. Model Capability

Table 1 presents the accuracy of LC-QAT compared against baseline VQ-QAT models, as well as SQ and VQ-based PTQ methods across multiple zero-shot QA tasks. As illustrated, our proposed LC-QAT achieves significantly higher accuracy than both the PTQ methods (SQ and VQ) and the VQ-QAT baseline, PV-Tuning. This performance gap demonstrates the superior capability of our model to leverage training data to recover quantization degradation.

A detailed analysis reveals that under the aggressive 2-bit quantization setting, SQ models suffer from substantial performance degradation, suggesting they are suboptimal starting points for high-performance QAT. In contrast, VQ-based models generally exhibit superior initial performance, validating the feasibility of using VQ as a foundation for subsequent optimization. Compared to AQLM, our LC-PTQ method using a linear-constrained codebook achieves higher initial performance. Furthermore, on the 8B model, LC-QAT performs slightly better than trellis-based methods such as QTIP and YAQA, while on the 1.7B model, which presents a greater compression challenge, our method brings significantly higher performance improvement after training, surpassing state-of-the-art PTQ/QAT methods.

Figure 4 shows the superior model capacity of our approach. As observed, the PV-Tuning method exhibits a performance plateau in the later stage of training, suggesting that the model reaches saturation within the initial steps. This limitation may stem from its asynchronous update strategy. In contrast, our method achieves consistent performance improvements and superior final accuracy as the volume of training data increases. This trend demonstrates the robust scaling capability of our framework, confirming that LC-QAT can effectively internalize more information from larger datasets.

*Table 1.* Comparison of the perplexity (PPL) on the Wikitext-2 test set and zero-shot QA accuracy between LC-QAT and other baselines. Under the same amount of training data, LC-QAT consistently outperforms vector quantization–aware training baselines. AE, AC, BQ, HS, WG, and PQ denote ARC-Easy, ARC-Challenge, BoolQ, HellaSwag, WinoGrande, and PIQA, respectively. Avg stands for Average, and Per stands for Performance. AQLM is under the commonly-used 2*8 configuration.

| MODEL | TYPE | METHOD | PPL↓ | AC↑ | AE↑ | BQ↑ | HS↑ | PQ↑ | WG↑ | AVG.↑ | PER.↑ |
|---|---|---|---|---|---|---|---|---|---|---|---|
| QWEN-3-8B | N/A | FP16 | 9.72 | 56.40 | 80.89 | 86.64 | 74.96 | 77.48 | 68.35 | 74.12 | 1.00 |
| | SQ | GPTQ | 4.68E4 | 26.79 | 25.67 | 42.87 | 25.84 | 52.50 | 50.04 | 37.29 | 0.50 |
| | | QUIP | 27.61 | 24.91 | 31.69 | 54.89 | 41.41 | 59.90 | 50.12 | 43.82 | 0.59 |
| | | OSTQUANT | 21.33 | 34.64 | 60.02 | 73.21 | 50.29 | 68.50 | 58.17 | 57.47 | 0.78 |
| | VQ | QUIP# | 12.38 | 46.50 | 68.43 | 83.12 | 66.62 | 74.32 | 66.30 | 67.55 | 0.91 |
| | | QTIP | 10.39 | 52.82 | 77.31 | 85.24 | 70.85 | 77.26 | 68.75 | 72.04 | 0.97 |
| | | YAQA | 10.50 | 53.16 | 78.41 | 83.27 | 71.41 | 77.42 | 69.14 | 72.14 | 0.97 |
| | | AQLM | 18.26 | 45.22 | 72.31 | 73.73 | 60.99 | 73.07 | 64.33 | 64.94 | 0.88 |
| | | LC-PTQ | 14.95 | 45.82 | 72.35 | 85.07 | 63.18 | 74.16 | 67.09 | 67.95 | 0.92 |
| | VQ-QAT | PV-TUNING | 10.65 | 51.79 | 75.46 | **83.82** | 70.44 | 76.88 | 67.56 | 70.99 | 0.96 |
| | | LC-QAT | **10.23** | **53.75** | **78.82** | 82.29 | **71.37** | **76.99** | **69.85** | **72.18** | **0.97** |
| QWEN-3-1.7B | N/A | FP16 | 16.72 | 43.08 | 69.69 | 77.52 | 60.37 | 72.14 | 61.80 | 64.10 | 1.00 |
| | SQ | GPTQ | 2.80E4 | 25.51 | 25.88 | 45.38 | 25.49 | 50.60 | 48.86 | 36.95 | 0.58 |
| | | QUIP | 133.16 | 24.06 | 32.45 | 46.97 | 30.92 | 52.61 | 49.25 | 39.38 | 0.61 |
| | | OSTQUANT | 149.80 | 25.68 | 27.86 | 61.95 | 28.13 | 52.56 | 41.37 | 39.59 | 0.62 |
| | VQ | QUIP# | 26.17 | 29.86 | 46.42 | 61.22 | 57.47 | 66.87 | 57.14 | 53.16 | 0.83 |
| | | AQLM | 40.25 | 28.24 | 47.72 | 69.48 | 42.42 | 63.43 | 56.03 | 51.22 | 0.80 |
| | | LC-PTQ | 29.11 | 31.91 | 53.87 | 67.46 | 44.99 | 61.80 | 54.38 | 52.40 | 0.82 |
| | | QTIP | 18.21 | 36.09 | 59.13 | 76.39 | 53.36 | 69.64 | 58.96 | 58.93 | 0.92 |
| | | YAQA | 17.14 | 37.03 | 61.99 | 78.41 | 54.83 | 70.24 | 59.27 | 60.30 | 0.94 |
| | VQ-QAT | PV-TUNING | 17.43 | 38.99 | 66.08 | 58.78 | 53.79 | 71.06 | 59.59 | 58.05 | 0.91 |
| | | LC-QAT | **13.44** | **41.72** | **68.18** | **69.85** | **58.23** | **72.69** | **57.93** | **61.43** | **0.96** |

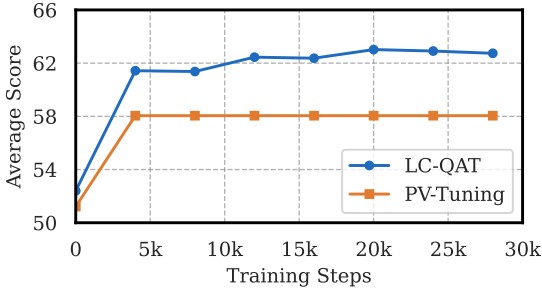

*Figure 4.* Average zero-shot task performance over training steps. LC-QAT steadily improves, while PV-Tuning saturates quickly.

## 5.2. Data Efficiency

Table 2 presents the zero-shot QA performance of LC-QAT and other baselines. Although LLM-QAT and EfficientQAT are trained with relatively smaller amounts of data, their performance remains substantially inferior. In contrast, our method surpasses ParetoQ while using only around 10% of its training data. Moreover, on several tasks, LC-QAT even exceeds the FP16 baseline. These results strongly validate our choice of vector quantization as the initialization for

QAT, which significantly reduces the required training data and improves the efficiency of quantization-aware training, thereby demonstrating the effectiveness of our approach in recovering performance under low-bit quantization.

Table 3 reports a comparison with BitNet. Our model achieves higher average accuracy than BitNet 2B4T and outperforms it on most tasks. These results indicate that our approach generalizes well across both reasoning and code benchmarks. Notably, while BitNet 2B4T attains strong performance using 4T tokens of training data, our method requires only approximately 0.1% of that data for fine-tuning to achieve comparable performance, further corroborating the substantial data efficiency advantage of our approach.

## 5.3. Ablation Studies

### 5.3.1. EFFECT OF PTQ INITIALIZATION

We first ablate the role of the PTQ starting point. Specifically, we replace the LC-PTQ initialization with random and GPTQ initialization, which leads to a severe degradation and fails to provide a usable starting point for subsequent training. Table 4 confirms that a high-quality PTQ initialization is essential for data-efficient 2-bit QAT. Experiments are performed on a Qwen3-8B model.

*Table 2.* Comparison of the perplexity (PPL) on the Wikitext-2 test set and zero-shot QA accuracy between LC-QAT and other baselines. Our method outperforms the current state-of-the-art ParetoQ, achieving higher performance using only around 10% of the training data. Results of all the baselines are from the evaluation of ParetoQ.

| MODEL | TYPE | METHOD | #TOKS | PPL↓ | AC↑ | AE↑ | BQ↑ | HS↑ | PQ↑ | WG↑ | AVG.↑ | PER.↑ |
|---|---|---|---|---|---|---|---|---|---|---|---|---|
| | N/A | FP16 | - | 6.2 | 57.7 | 81 | 83.6 | 79.5 | 81 | 73.9 | 76.12 | 1.00 |
| | | RTN | | 1.20E6 | 25.10 | 27.20 | 37.80 | 26.10 | 49.70 | 50.50 | 36.07 | 0.47 |
| | | GPTQ | | 160.00 | 26.10 | 27.00 | 61.60 | 26.00 | 50.50 | 49.70 | 40.15 | 0.53 |
| | PTQ | AWQ | N/A | 1.10E6 | 27.10 | 26.00 | 58.30 | 26.10 | 51.40 | 49.80 | 39.78 | 0.52 |
| | | OMNIQ | | 7.60E4 | 22.80 | 27.30 | 37.90 | 25.30 | 49.50 | 49.40 | 35.37 | 0.46 |
| LLAMA-3-8B | | SPINQUANT | | 31.20 | 22.00 | 32.40 | 59.00 | 31.90 | 53.20 | 49.90 | 41.40 | 0.54 |
| | | LLM-QAT | 100M | 29.50 | 35.90 | 54.80 | 64.80 | 58.00 | 68.00 | 54.70 | 56.03 | 0.74 |
| | QAT | EFFICIENTQAT | 24M | 9.60 | 46.80 | 69.30 | 75.05 | 69.00 | 76.40 | 66.30 | 67.14 | 0.88 |
| | | PARETOQ | 30B | **8.00** | 54.50 | 78.50 | 76.40 | 73.80 | **79.20** | **70.00** | 72.07 | 0.95 |
| | | LC-QAT | **4B** | 9.38 | **57.94** | **82.95** | **77.46** | **76.67** | 78.62 | 66.85 | **73.42** | **0.96** |
| | N/A | FP16 | - | 7.70 | 50.70 | 72.60 | 74.60 | 74.30 | 78.20 | 69.20 | 69.93 | 1.00 |
| | | RTN | | 7.80E5 | 25.10 | 26.90 | 37.80 | 25.70 | 50.10 | 49.60 | 35.87 | 0.51 |
| | | GPTQ | | 270.00 | 22.90 | 28.60 | 46.40 | 27.10 | 50.00 | 50.10 | 37.52 | 0.54 |
| LLAMA-3-3B | PTQ | AWQ | N/A | 6.20E5 | 27.50 | 27.30 | 38.20 | 26.10 | 51.10 | 50.70 | 36.82 | 0.53 |
| | | OMNIQ | | 6.50E3 | 24.60 | 28.30 | 37.80 | 25.30 | 50.50 | 50.20 | 36.12 | 0.52 |
| | | SPINQUANT | | 57.40 | 23.70 | 28.30 | 53.20 | 26.10 | 51.10 | 49.00 | 38.57 | 0.55 |
| | | LLM-QAT | 100M | 2.90E5 | 33.30 | 49.30 | 63.50 | 48.90 | 65.20 | 52.20 | 52.07 | 0.74 |
| | QAT | PARETOQ | 30B | **9.10** | 49.00 | 73.09 | **68.80** | 69.20 | **76.40** | **64.40** | 66.82 | 0.96 |
| | | LC-QAT | **4B** | 13.20 | **52.39** | **78.75** | 66.51 | **69.83** | 76.22 | 66.30 | **68.33** | **0.98** |

*Table 3.* Performance comparison with BitNet 2B4T on math and code tasks. Our method achieves higher performance than BitNet 2B4T on most tasks and also surpasses it in the average score.

| METRIC | BITNET 2B | LC-QAT 1.7B |
|---|---|---|
| OPENBOOKQA | 41.60 | **55.20** |
| IFEVAL | 53.48 | **58.63** |
| MMLU | **53.17** | 46.77 |
| GSM8K | 58.38 | **58.61** |
| MATH | **43.40** | 39.60 |
| HUMANEVAL | 38.40 | **43.29** |
| AVG. | 48.07 | **50.35** |
| #TOKENS | 4T | **4B** |

### 5.3.2. VALUE OF END-TO-END TRAINING

We then evaluate whether the gains of LC-QAT come from the end-to-end training rather than a better initialization alone. We run PV-Tuning's coordinate-descent optimization starting from the same LC-PTQ initialization used by LC-QAT. As shown in Table 5, although PV-Tuning improves upon LC-PTQ, LC-QAT achieves substantially higher accuracy under the same initialization.

### 5.3.3. ABLATIONS OF AUXILIARY COMPONENTS

**Differentiable gradient estimator.** As Figure 5b shows, our ablation experiments demonstrate that using the differentiable gradient estimator significantly mitigates the initial loss spike observed when using the standard STE. Without DGE, the model exhibits a pronounced loss spike during early training steps, while incorporating the differentiable gradient estimator stabilizes the loss curve, allowing the model to descend smoothly from the start and maintain

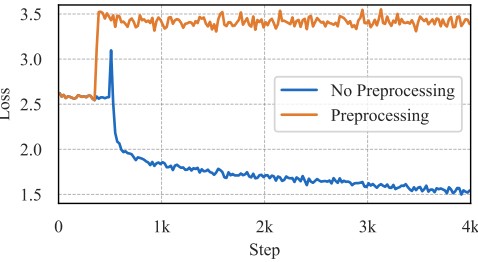

*(a)* Ablation study on integer weight preprocessing

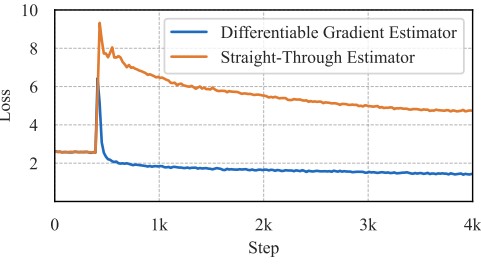

*(b)* Ablation study on gradient estimation

*Figure 5.* (a) Without preprocessing, the training loss remains nearly constant. With preprocessing, the loss decreases continuously, demonstrating that aligning integer weights with a Xavier-initialized distribution is essential for stable training and effective gradient propagation. (b) When using the STE, the spikes are extremely large and difficult to recover. In contrast, using the DGE results in significantly smaller spikes and more stable convergence.

consistent learning throughout training. This highlights the effectiveness of DGE in ensuring robust gradient flow and reliable optimization under low-bit settings.

*Table 4.* Ablation on the PTQ initialization. Replacing the LDLQ-based initialization with random initialization severely degrades the zero-shot QA accuracy.

| INIT. METHOD | AC↑ | AE↑ | BQ↑ | HS↑ | PQ↑ | WG↑ | AVG.↑ | PER.↑ |
|---|---|---|---|---|---|---|---|---|
| LC-PTQ | **45.82** | **72.35** | **85.07** | **63.18** | **74.16** | **67.09** | **67.95** | **0.92** |
| GPTQ INIT. | 19.62 | 32.49 | 39.72 | 27.28 | 54.79 | 51.30 | 37.53 | 0.51 |
| RANDOM INIT. | 27.13 | 25.55 | 56.88 | 26.22 | 51.69 | 49.57 | 39.51 | 0.54 |

*Table 5.* Comparison of PV-Tuning and end-to-end training of a Qwen-3-1.7B LC-PTQ model. LC-QAT consistently achieves higher accuracy, demonstrating the value of end-to-end training.

| METHOD | AC↑ | AE↑ | BQ↑ | HS↑ | PQ↑ | WG↑ | AVG.↑ |
|---|---|---|---|---|---|---|---|
| LC-PTQ | 31.91 | 53.87 | 67.46 | 44.99 | 61.80 | 54.38 | 52.40 |
| LC-PTQ + PV-TUNING | 36.69 | 61.95 | 57.43 | 49.95 | 69.31 | 56.75 | 55.35 |
| LC-QAT | **39.51** | **64.35** | **70.43** | **55.01** | **71.65** | **57.54** | **59.75** |

**Integer Weight Preprocessing.** We conduct an ablation study to assess the role of integer weight preprocessing in LC-QAT. As shown in Figure 5a, removing this preprocessing prevents effective training. This demonstrates that integer weight preprocessing is a necessary component for the stable training of LC-QAT.

### 5.3.4. SENSITIVITY AND SCALING ANALYSES

We find that LC-QAT continues to improve as the training budget increases from 1B to 10B tokens, which indicates that LC-QAT can benefit from continuous scaling and achieve higher performance compared to the 4B token training used in Table 1 (Appendix A.5.1, Table 10). Furthermore, we extend our experiments to 14B models and observe that LC-QAT still achieves notable performance gains at this larger scale (Appendix A.5.2, Table 11).

We also find that the PTQ initialization is insensitive to the dimension $d$ of the matrix $A$ ($d = 4$ vs. $d = 8$), and given that a larger $d$ introduces more additional parameters, we consider $d = 4$ to be a suitable choice (Appendix A.4). Additionally, the calibration data may affect the outcome of PTQ initialization; we find that using the commonly adopted RedPajama dataset yields favorable results on zero-shot QA tasks (Appendix A.4).

## 6. Conclusion

We presented LC-QAT, a novel framework for end-to-end training of 2-bit VQ models. By introducing linear-constrained codebooks, LC-QAT enables synchronized optimization of discrete and continuous parameters.

Combined with a strong post-training initialization and smooth gradient estimation, LC-QAT achieves superior accuracy and data efficiency compared with existing scalar- and vector-quantized QAT methods. Extensive experiments demonstrate that LC-QAT exhibits stronger model capa-

bility than conventional VQ-QAT methods, and achieves competitive performance using significantly less training data compared to scalar-quantization-based QAT methods.

Our work offers another option for 2-bit QAT with higher data-efficiency and superior performance. The future work will focus on the scalability of LC-QAT and the optimization of its inference performance.

## Limitations

Our experiments primarily focus on data-efficient fine-tuning under limited training budgets, and the scalability to large-scale training remains an open question.

Moreover, LC-QAT introduces an additional linear projection in the forward pass. Although this operation has limited complexity, its practical interaction with highly optimized GEMM kernels in modern deep learning frameworks has not been systematically studied. The impact of this decoding process on training efficiency and decoding throughput under different conditions remains to be fully explored.

## Acknowledgements

This work is supported by the National Key Research and Development Program of China (2024YFB4505603) and the National Natural Science Foundation of China (No. 62576186). This work is also supported by Tsinghua KA Excellence Center. This work is partially supported by Tsinghua University (Department of Computer Science and Technology) - Sinopec Joint Research Center for Artificial Intelligence.

## Impact Statement

This paper presents work whose goal is to advance the field of Machine Learning. There are many potential societal consequences of our work, none which we feel must be specifically highlighted here.

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

# A. Appendix

## A.1. Samples of Training Data

This section mainly presents example samples from the two datasets used for training: FineWeb and AM-Qwen3-Distilled. As Figure 6 shows, FineWeb is a large-scale, high-quality web text dataset curated for instruction-tuning and general language understanding tasks. As Figure 7 shows, AM-Qwen3-Distilled is a distilled dataset generated from the outputs of Qwen3 models, designed to provide high-quality supervised data for improving instruction-following and reasoning capabilities.All training is conducted with a sequence length of 2048.

---

**Sample1:**
The Net Neutrality repeal vote is coming. Tell these Dems to vote Yes. The House of Representatives is likely to vote tomorrow, Thursday, on the repeal of the FCC's Net Neutrality power grab. Using the Congressional Review Act, the repeal of the Net Neutrality order can be accomplished in an expedited way. In particular this means the bill cannot be filibustered in the Senate, so passing it means something. As Seton Motley said: This is our first opportunity . Read More.

**Sample2:**
Tips for Preventing Medicare Fraud: The Department of Health and Human Services (HHS) Office of the Inspector General (OIG) has created a new web site to provide tips for preventing Medicare fraud and medical identity theft. See HHS news release. OIG's new web site includes a brochure containing tips, where to report fraud, and other resources. In addition, CMS issued a Medicare Fraud & Abuse Fact Sheet earlier this year.

**Sample3:**
Great decorating addition.I have a grape/Italian theme in my kitchen. I purchased 5 of these. I decided to use them to put around my pull knobs on my overhead cabinets. Now I am ordering more to sprinkle around in other places in the kitchen - even to hang up via suction cups on my white kitchen tile. September 20, 2012.

---

*Figure 6.* Examples of FineWeb.

---

**Sample1:**
  **human:** Write a python function to reverse the strings in a given list of strings. For example, given the list ["hello", "world"], the function should return ["olleh", "dlrow"].
  **assistant:** python def reverse_strings(list_of_strings): return [s[::-1] for s in list_of_strings]
**Sample2:**
  **human:** Write a python function that takes in two integers, a and b, and returns the sum of the two integers.
  **assistant:** python def add_numbers(a, b): return a + b
**Sample3:**
  **human:** I need to create a Python function that converts a given string to uppercase. How can I achieve this?
  **assistant:** python def to_uppercase(input_string): return input_string.upper()

---

*Figure 7.* Examples of AM-Qwen3-Distilled showing human instructions and assistant responses.

## A.2. Inference Speed

We report inference throughput on a single NVIDIA A100 GPU with batch size 1 and sequence length 1024 (CUDA Graph enabled). As shown in Table 6, LC-QAT achieves the highest throughput among VQ baselines while supporting both $d = 4$ and $d = 8$ with nearly identical speed.

*Table 6.* Throughput comparison on LLaMA-3-8B (single A100, bs=1, 1024 tokens, CUDA Graph).

| METHOD | $d$ | THROUGHPUT (TOK/S) ↑ |
|---|---|---|
| FP16 | – | 60.38 |
| AQLM 2×8 | 8 | 70.97 |
| QuIP# E8P | 8 | 79.70 |
| QuIP# D4 | 4 | 23.80 |
| LC-QAT | 4 | **101.65** |
| LC-QAT 2×8 | 8 | 101.25 |

The inference throughput in Table 6 is measured on a single NVIDIA A100 GPU with batch size 1 and sequence length

1024 (CUDA Graph enabled). Our implementation consists of (1) the Hadamard transformation using the publicly available fast-hadamard-transform library, and (2) a custom fused CUDA kernel implementing the reformulated forward pass in Equation 8, which fuses affine dequantization ($A \cdot w_{int} + B$), dot-product accumulation, and block reduction into a single kernel launch to minimize memory traffic.

Currently, only QuIP# provides an official CUDA kernel supporting $d = 4$ but it is not well-optimized for bs=1 inference, and AQLM/QTIP hardcode $d$ in their kernels. In contrast, LC-QAT's affine dequantization is structurally simple and generalizes across group sizes with the same fused kernel, which explains the strong throughput and the negligible overhead when increasing $d$.

*Table 7.* Total wall-clock time comparison including PTQ initialization (estimated on 8 A800 GPUs).

| METHOD | PTQ TIME (H) | QAT TIME (H) | TOTAL TIME (H) |
|---|---|---|---|
| LC-QAT | 6 | 55 | 61 |
| PARETOQ | N/A | 417 | 417 |

## A.3. Detailed Results of Preliminary Optimization Analysis

Table 8 shows the performance discrepancy between the initialization point used by LC-QAT and that of scalar quantization. In this experiment, we select OSTQuant (Xing et al., 2025), which is a state-of-the-art SQ method, as the baseline. Similar to our approach, OSTQuant utilizes the Hadamard transform to mitigate the impact of outliers.

While OSTQuant performs excellently at 4 bits and above, the results in Table 8 reveal that it suffers from a near 40% performance loss under an aggressive 2-bit quantization strategy. Notably, on more complex tasks such as ARC-C, its performance degrades to a level close to random guessing. In contrast, the LC-QAT initialization preserves over 84% of the zero-shot QA accuracy. These findings, consistent with the results in Section 2.3, validate the superior quality of our initialization. Given that OSTQuant also incorporates an online Hadamard transform, the only distinction between the two lies in the choice of vector quantization versus scalar strategies. This comparison underscores the exceptional ability of VQ to maintain model performance in the ultra-low bit-width regime.

*Table 8.* Zero-shot performance comparison on LLaMA3 models. We report the initial point of LC-QAT (LC-PTQ) and scalar quantization baselines on zero-shot commonsense reasoning benchmarks.

| MODEL | METHOD | AC | AE | BQ | HS | PQ | WG | AVG. | PER. |
|---|---|---|---|---|---|---|---|---|---|
| | FP16 | 50.07 | 72.60 | 74.60 | 74.30 | 78.20 | 69.20 | 69.83 | 1.00 |
| LLaMA3-3B | OSTQUANT | 23.81 | 34.34 | 59.51 | 33.23 | 55.82 | 50.99 | 42.95 | 0.62 |
| | LC-PTQ | 33.28 | 58.08 | 69.14 | 57.96 | 71.06 | 60.85 | 58.40 | 0.84 |
| | FP16 | 56.57 | 80.93 | 86.61 | 74.94 | 77.80 | 67.88 | 74.12 | 1.00 |
| LLaMA3-8B | OSTQUANT | 25.17 | 39.27 | 60.67 | 37.79 | 60.39 | 52.41 | 42.95 | 0.58 |
| | LC-PTQ | 38.65 | 66.04 | 74.74 | 67.02 | 75.90 | 65.59 | 64.66 | 0.87 |

## A.4. Sensitivity to Group Size

We study the sensitivity of LC-PTQ to the VQ group size $d$ and calibration data. On Qwen-3-1.7B at the PTQ stage, changing the group size from $d = 4$ to $d = 8$ results in only marginal differences in zero-shot accuracy, while switching the calibration dataset from RedPajama to an in-house dataset introduces a modest PTQ-level gap.

## A.5. Scaling Analysis

### A.5.1. DATA SCALING

We conduct a data-scaling analysis on Qwen-3-1.7B, where performance continues to improve beyond 4B tokens without saturation. This supports our primary claim of *data efficiency*: LC-QAT achieves competitive downstream accuracy with a substantially smaller data budget, while a symmetric scaling analysis for ParetoQ is not feasible since its training data is not publicly available.

*Table 9.* Sensitivity of LC-PTQ to group size $d$ and calibration data (Qwen-3-1.7B, PTQ stage).

| $d$ | CALIBRATION | ARC-C | ARC-E | BOOLQ | HELLASWAG | PIQA | WINOGRANDE | AVG. |
|---|---|---|---|---|---|---|---|---|
| 4 | REDPAJAMA | 29.35 | 44.87 | 63.79 | 43.88 | 64.96 | 54.46 | 50.22 |
| 8 | REDPAJAMA | 29.10 | 45.66 | 63.49 | 43.77 | 64.85 | 55.33 | 50.37 |
| 4 | IN-HOUSE | 27.47 | 43.06 | 62.26 | 37.71 | 62.46 | 54.78 | 47.96 |

*Table 10.* Data scaling analysis on Qwen-3-1.7B.

| #TOKENS | ARC-C | ARC-E | BOOLQ | HELLASWAG | PIQA | WINOGRANDE | AVG. |
|---|---|---|---|---|---|---|---|
| 1B | 39.51 | 64.35 | 70.43 | 55.01 | 71.65 | 57.54 | 59.75 |
| 4B | 41.72 | 68.18 | 69.85 | 58.23 | 72.69 | 57.93 | 61.43 |
| 10B | 42.41 | 68.90 | 69.02 | 59.60 | 73.45 | 61.33 | 62.45 |

### A.5.2. PARAMETER SCALING

We further evaluate LC-QAT on a 14B-parameter model using approximately 58M tokens. The advantage of LC-QAT over the PTQ initialization remains consistent at this larger scale.

*Table 11.* Results on a 14B model (58M tokens).

| METHOD | WIKI | C4 | ARC-C | ARC-E | BOOLQ | HELLASWAG | PIQA | WINOGRANDE | AVG. | PER. |
|---|---|---|---|---|---|---|---|---|---|---|
| FP16 | 8.64 | 13.81 | 60.15 | 82.83 | 89.30 | 78.84 | 79.87 | 72.85 | 77.31 | 1.00 |
| LC-PTQ | 11.41 | 16.85 | 51.62 | 77.69 | 87.16 | 69.82 | 77.42 | 71.59 | 72.55 | 0.94 |
| LC-QAT | 10.08 | 15.79 | 54.69 | 79.25 | 85.90 | 73.66 | 78.51 | 73.88 | 74.32 | 0.96 |

## A.6. Additional Reasoning Benchmarks of Base Models

We evaluate LC-QAT and PV-Tuning on harder reasoning benchmarks. LC-QAT consistently achieves higher accuracy.

*Table 12.* Additional reasoning benchmarks.

| TASK | FP16 | LC-QAT | PV-TUNING |
|---|---|---|---|
| MMLU | 55.26 | 45.92 | 44.70 |
| CEVAL | 58.25 | 36.03 | 34.70 |
| CMMLU | 56.95 | 36.46 | 35.56 |

