# OpenReview forum: "LC-QAT: Data-Efficient 2-Bit QAT for LLMs via Linear-Constrained Vector Quantization"
_ICML.cc/2026/Conference — ICML 2026 regular_

### Official Review · Reviewer_Lu6C · 2026-03-03

**Soundness:** 2
**Presentation:** 2
**Significance:** 3
**Originality:** 2
**Overall Recommendation:** 3
**Confidence:** 3

**Summary:**

The authors propose LC-QAT. It is a 2-bit quantization-aware training framework for large language models. The method uses a linear-constrained codebook. This codebook replaces the discrete lookup process found in standard vector quantization. The framework assembles several existing techniques. It uses Hadamard transforms and LDLQ for post-training quantization initialization. It implements an affine mapping to allow gradients to flow during the forward pass. It also incorporates a previously published Differentiable Gradient Estimator. The experiments on LLaMA-3 models indicate that the method recovers performance using very little training data.

**Compliance With Llm Reviewing Policy:**

Affirmed.

**Final Justification:**

I have improved my score.

**Key Questions For Authors:**

* How do you isolate the contribution of your specific affine mapping from the existing techniques used in your pipeline? Is the method strictly an incremental combination?
* How does the linear-constrained codebook maintain its discrete integrity during continuous parameter updates? Please provide the missing theoretical details that were deferred to the concurrent work.
* Can you provide a detailed ablation study? This study should separate the performance gain of the LC-PTQ initialization from the actual QAT fine-tuning phase. This would clarify what truly drives the data efficiency.
* What is the actual end-to-end memory bandwidth overhead for the chunk-wise projection $X_i \mathbf{A}$ during inference? Please provide concrete profiling results on target hardware rather than theoretical complexity estimations.

**Limitations:**

Yes. The authors mention the lack of large-scale training validation and the potential inference overhead from the added linear projection. However, they fail to address the severe algorithmic limitation. They rely entirely on an unpublished concurrent work for their core methodology.

**Strengths And Weaknesses:**

**Strengths:**
* The submission tackles 2-bit weight quantization. This addresses a critical efficiency bottleneck for edge deployment scenarios.
* The empirical results display excellent data efficiency. The model achieves competitive accuracy using only 0.1% to 10% of standard training data budgets.
* The forward pass reformulation is clever. Computing $Y = \sum_{i=1}^{G} (X_i A) W_{z,i}^T + \sum_{i=1}^{G} (X_i B) \mathbf{1}^T$ allows the framework to reuse optimized integer matrix multiplication kernels.

**Weaknesses:**
* The submission completely lacks self-containment. The authors defer the core vector quantization details to an unpublished paper. They state, "Further details regarding the vector quantization using the linear-constrained codebook are provided in our concurrent work" (Page 3, Lines 287-288). I cannot evaluate the mathematical validity of the codebook initialization without this reference.
* The method appears to be a straightforward incremental assembly. The pipeline strings together LDLQ, Hadamard transforms, a basic affine mapping, and a Differentiable Gradient Estimator. The boundaries of the actual novel theoretical contribution are blurry.
* Model compression indicates a clear trend. The success of extreme low-bit QAT depends almost entirely on the PTQ initialization quality. The paper claims the QAT framework drives the data efficiency. However, the heavy lifting seems to come from the hidden LC-PTQ initialization rather than the continuous training mechanics.
* The paper lacks hardware-aware latency profiling. The authors claim the overhead of the linear projection is minimal. However, mapping integer indices through floating-point matrices often introduces memory bandwidth bottlenecks on actual hardware. The authors leave this critical evaluation to future work.

---

> ### Author Rebuttal · Authors · 2026-03-31
>
> We thank the reviewer for the detailed feedback. We address each concern below.
>
> **W1 & Q2: Self-containment and discrete integrity during training.**
>
> We respectfully clarify that LC-QAT is fully self-contained. The concurrent work cited on Line 287 describes our PTQ method in greater detail, but **all components necessary to understand and reproduce LC-QAT are present in this submission**: the linear-constrained codebook parameterization (Eq. 3–4), the LDLQ-based initialization, the forward/backward pass (Section 3), and the integer weight preprocessing (Section 3.4). The mathematical validity of our codebook follows directly from standard properties of affine mappings over integer grids — analogous to QuIP# and NestQuant, but with the key distinction that our construction admits a single shared linear transformation enabling end-to-end gradient flow. The concurrent work was provided as supplementary material alongside this paper for additional reference. We will further strengthen the self-contained description in the revision.
>
> Regarding discrete integrity: as we mentioned in Eq. 5, during training, LC-QAT maintains continuous proxy weights $W_p$, but every forward pass applies round + clamp to produce integer weights $W_z \in {0,1,2,3}^{m\times n}$. This ensures the codebook's discrete structure is enforced at every training step. The proxy weights are updated by gradients, but the actual weights used for computation are always valid discrete vectors — the same mechanism used by all SQ-QAT methods (e.g., ParetoQ, EfficientQAT).
>
> **W2 & Q1: Novelty — incremental assembly of existing components.**
>
> We respectfully disagree. The core contribution is enabling, for the first time, **fully differentiable end-to-end training of vector-quantized LLMs**. Prior VQ methods are fundamentally limited to PTQ because discrete codebook lookup is non-differentiable, and the only existing VQ-QAT attempt (PV-Tuning) resorts to expensive, asynchronous coordinate descent that cannot scale.
>
> Our key insight — replacing unconstrained codebook lookup with a linear-constrained parameterization that reduces index selection to rounding and clamping — is what unlocks standard backpropagation for VQ. This is not merely "stringing together" existing components: LDLQ, Hadamard transforms, and DGE are engineering choices that stabilize the training process, just as Adam and learning rate warmup are not the contribution of other QAT papers. The resulting framework achieves near-complete performance recovery under 2-bit quantization with orders-of-magnitude less training data than SQ-QAT — an outcome no prior combination of these components has achieved.
>
> **W3 & Q3: Performance driven by LC-PTQ initialization rather than QAT.**
>
> We agree that high-quality initialization is a key enabler — and we explicitly claim this. However, two points are important:
>
> First, the strong initialization is only possible *because* our linear-constrained codebook supports both PTQ and QAT within the same parameterization. The data efficiency is an emergent property of the unified framework, not either component alone.
>
> Second, QAT provides substantial gains beyond PTQ. From Table 1 (Qwen-3-1.7B): LC-PTQ achieves 52.40 Avg. (0.82 Per.), while LC-QAT reaches **61.43 Avg. (0.96 Per.)** — recovering nearly all FP16 performance. On Qwen-3-8B: LC-PTQ = 67.95 (0.92 Per.) → LC-QAT = 72.18 (0.97 Per.). Figure 4 further shows that LC-QAT continues improving steadily with more data, confirming that end-to-end training contributes meaningfully beyond initialization.
>
> Additionally, we ran PV-Tuning from the same LC-PTQ initialization. It achieves only 55.35 Avg. vs. LC-QAT's 59.75 on Qwen-3-1.7B (details in our response to Reviewer LGNi), confirming that the training mechanics — not just initialization — drive the advantage.
>
> **W4 & Q4: Hardware-aware latency and memory bandwidth.**
>
> We wish to clarify a misconception: LC-QAT's inference does **not** map integer indices through floating-point matrices. Through the reformulation in Eq. (8), the forward pass computes $Y = Σ(X_i A)W^T_{z,i} + Σ(X_i B)1^T$ — the affine correction is absorbed into the activations ($X_i A$), while $W_z$ remains an integer matrix.
>
> For hardware computation details,  we implement a custom fused CUDA kernel for this computation. Since $W_z$ remains in low-bit integer format, the memory bandwidth requirement is comparable to standard SQ inference. As reported in Appendix A.2, LC-QAT achieves **101.65 tok/s** on Llama-3-8B (single A100), compared to 60.38 for FP16 and 70.97 for AQLM — a **1.68× speedup** over FP16 with over 75% memory reduction. This demonstrates that our method incurs no meaningful latency overhead in practice.

---

> > ### Author Rebuttal · Reviewer_Lu6C · 2026-04-01
> >
> > Thank you for the detailed reply, I will increase my score.

---

> > > ### Author Response · Authors · 2026-04-05
> > >
> > > We thank the reviewer for acknowledging that the concerns have been **fully resolved**. We will incorporate all discussed improvements into the revision. Given that the concerns have been addressed, we kindly ask the reviewer to reconsider whether the current score fully reflects this assessment.

---

### Official Review · Reviewer_LGNi · 2026-03-03

**Soundness:** 3
**Presentation:** 3
**Significance:** 3
**Originality:** 3
**Overall Recommendation:** 4
**Confidence:** 4

**Summary:**

The paper proposes LC-QAT, a 2-bit weight-only quantization-aware training framework for large language models that replaces the conventional vector-quantization codebook lookup with a linear-constrained parameterization. This reparameterization enables a lookup-free forward pass with differentiable training via rounding/clamping on proxy weights and a differentiable gradient estimator. Experiments on Qwen-3 and LLaMA-3 (1.7B–8B) show improved accuracy and notably higher data efficiency compared to prior VQ-QAT (PV-Tuning) and 2-bit SQ-QAT (e.g., ParetoQ).

**Compliance With Llm Reviewing Policy:**

Affirmed.

**Final Justification:**

Most of my concerns have been addressed, and I appreciate the authors’ clarifications and additional experiments. I would like to lean towards accept.

**Key Questions For Authors:**

Please see weaknesses.

**Limitations:**

yes

**Strengths And Weaknesses:**

**Strengths**

1. Introduces a linear-constrained codebook (c = A z + B) that turns discrete VQ lookup into SQ-like discretization followed by a linear mapping, making end-to-end VQ-QAT feasible with standard backprop.

2. Addresses STE instability with a differentiable gradient estimator (DGE).

3. Empirical evidence of data efficiency is compelling (e.g., outperforming ParetoQ with an order of magnitude fewer tokens).

**Weaknesses**

1.  Fairness of some comparisons needs strengthening: PV-Tuning is initialized from AQLM, whereas LC-QAT uses a stronger LC-PTQ initialization; it would be informative to run PV-Tuning from LC-PTQ as well.

2. Limited analysis of sensitivity to group size $d$; no ablation on $A/B$ sharing granularity (per-layer vs per-block); no ablation of different calibration dataset.

3. While VQ-based methods offer a superior bit-performance trade-off, they often lack native hardware support, as modern GPU kernels for low-bit data are typically optimized for SQ formats. Please clarify the specific kernel implementation used for the benchmarks in Table 4 , whether it relies on custom CUDA code or standard libraries, and specify the exact hardware environment.

4. While the paper highlights data efficiency in terms of token count, the total training time must be transparently evaluated. Since the LC-QAT pipeline mandates a high-quality initialization derived from PTQ processes like LDLQ. Please compare the total wall-clock time with PARETOQ for the full pipeline, including the initialization phase.

---

> ### Author Rebuttal · Authors · 2026-03-31
>
> We thank the reviewer for the thorough and constructive feedback. We address each concern below.
>
> **W1: Fairness of comparison — PV-Tuning initialized from LC-PTQ.**
>
> To ensure a fair comparison, we implemented PV-Tuning's coordinate descent optimization starting from the same LC-PTQ initialization used by LC-QAT. Results on Qwen-3-1.7B (1B tokens):
>
> |                    | ARC-C | ARC-E | BoolQ | HellaSwag | PIQA  | WinoGrande | Avg.  |
> | ------------------ | ----- | ----- | ----- | --------- | ----- | ---------- | ----- |
> | LC-PTQ             | 31.91 | 53.87 | 67.46 | 44.99     | 61.80 | 54.38      | 52.40 |
> | LC-PTQ + PV-Tuning | 36.69 | 61.95 | 57.43 | 49.95     | 69.31 | 56.75      | 55.35 |
> | LC-QAT             | 39.51 | 64.35 | 70.43 | 55.01     | 71.65 | 57.54      | 59.75 |
>
> PV-Tuning from LC-PTQ performs worse than LC-QAT despite sharing the same starting point. This confirms that the performance advantage comes from LC-QAT's fully differentiable, synchronous end-to-end optimization, not solely from the stronger initialization. PV-Tuning's asynchronous coordinate descent limits its ability to exploit training data — as also evidenced by its early saturation in Figure 4.
>
> **W2: Sensitivity to group size d, A/B sharing granularity and different calibration dataset.**
>
> In a preliminary experiment on Qwen2.5-7B, we compared matrix-wise and model-wise sharing of A and B. While model-wise sharing leads to a noticeable LC-PTQ performance drop relative to matrix-wise sharing, approximately 1B tokens of QAT fine-tuning recovers this gap, with both configurations converging to comparable final accuracy. This demonstrates that LC-QAT's end-to-end training is robust to the choice of sharing granularity. We adopt matrix-wise sharing as the default as it provides a stronger PTQ starting point with negligible parameter overhead.
>
> |sharing granularity|	Ceval|	MMLU|	PIQA|	AVG.|
> | --------- | ------- | ------- | -------- |-------- |
> |N/A (FP16)|	74.29| 	65.61 |	72.09| 	61.02|
> |matrix wise|	56.09 |	49.67| 	68.50 |	58.09 |
> |finetuning 1B|	62.55| 	60.60| 	72.36| 	65.17|
> |model wise|	28.45| 	44.84| 	67.03| 	46.77 |
> |finetuning 1B|	62.18| 	60.00| 	73.67| 	65.28|
>
> Due to time constraints we were unable to run full QAT experiments across these variables, but we report PTQ-stage results on Qwen3-1.7B as indicative evidence. Varying group size between d=4 and d=8 yields only marginal differences in zero-shot accuracy (50.22 vs. 50.37 average). Similarly, switching the calibration dataset from RedPajama to an in-house dataset produces a small PTQ-level gap (50.22 vs. 47.96). Consistent with the sharing granularity findings above, we expect these already-small differences to diminish further after QAT fine-tuning.
>
> | group size | calibration dataset | ARC-C | ARC-E | BoolQ | HellaSwag | PIQA  | WinoGrande | Avg.  |
> | ---------- | ------------------- | ----- | ----- | ----- | --------- | ----- | ---------- | ----- |
> | 4          | pajama              | 29.35 | 44.87 | 63.79 | 43.88     | 64.96 | 54.46      | 50.22 |
> | 8          | pajama              | 29.10 | 45.66 | 63.49 | 43.77     | 64.85 | 55.33      | 50.37 |
> | 4          | in-house data       | 27.47 | 43.06 | 62.26 | 37.71     | 62.46 | 54.78      | 47.96 |
>
> **W3: Kernel implementation and hardware details for Table 4.**
>
> The inference throughput in Table 4 is measured on a single NVIDIA A100 GPU with batch size 1 and sequence length 1024. Our implementation consists of: (1) the Hadamard transformation using the publicly available fast-hadamard-transform library, and (2) a custom fused CUDA kernel implementing the reformulated forward pass in Eq. (8), which fuses dequantization (the affine mapping A·w_int + B), dot-product accumulation, and block reduction into a single kernel launch to minimize memory traffic. We will add full implementation details to the appendix.
>
> **W4: Total wall-clock time comparison including PTQ initialization.**
>
> We agree that total pipeline time should be transparently evaluated. The LC-QAT pipeline consists of two stages: (1) PTQ initialization via LDLQ, and (2) QAT fine-tuning. We provide the breakdown below:
>
> | Method  | PTQ time (hours) | QAT time (hours) | total time (hours) |
> | ------  | ---------------- | ---------------- | ------------------ |
> | LC-QAT  | 6                |               55 |                  61|
> | ParetoQ | N/A              |              417 |                 417|
>
>
> The training time was estimated on 8 A800 GPUs. We note that LC-QAT's PTQ initialization is a one-time offline cost shared with all VQ-PTQ methods (QuIP#, AQLM, etc.). The QAT stage itself is efficient due to the data-efficient nature of our method — training on only 4B tokens takes significantly less time than ParetoQ's 30B-token training.

---

> > ### Author Rebuttal · Reviewer_LGNi · 2026-04-01
> >
> > Most of my concerns have been addressed, and I appreciate the authors’ clarifications and additional experiments.
> >
> > That said, I would still like to see some of these details further strengthened in the final version.
> >
> > In particular, for VQ-based methods, dequantization is often a key bottleneck and is highly sensitive to the choice of group size. The codebook-based dequantization in VQ introduces additional memory access and cache overhead. Prior work (e.g., QTIP at 2-bit) shows around ~20% higher latency compared to LUT-based GEMM kernels (ICLR’24), suggesting that strong perplexity performance does not necessarily translate to better end-to-end efficiency.
> >
> > While the paper provides a high-level description of the kernel implementation, I encourage the authors to include a more detailed comparison with existing VQ kernels under the same group size setting in the revision. This would help better contextualize the practical efficiency gains of the proposed method.
> >
> > Overall, I maintain my positive assessment of the paper. Good luck.

---

> > > ### Author Response · Authors · 2026-04-05
> > >
> > > We thank the reviewer for acknowledging our responses and for the constructive suggestion. We have conducted a detailed kernel-level investigation as recommended.
> > >
> > > **Throughput comparison (Llama-3-8B, single A100, bs=1, 1024 tokens, CUDA Graph):**
> > >
> > > | Method | d | Throughput (tok/s) |
> > > | ------ | - | ------------------ |
> > > | FP16 | — | 60.38 |
> > > | AQLM 2×8 | 8 | 70.97 |
> > > | QuIP# E8P | 8 | 79.70 |
> > > | QuIP# D4 | 4 | 23.80 |
> > > | LC-QAT | 4 | 101.65 |
> > > | LC-QAT 2×8 | 8 | 101.25 |
> > >
> > > Currently, only QuIP# provides an official CUDA kernel supporting group size d=4, but it is not well-optimized for bs=1 inference. AQLM and QTIP hardcode the group size in their CUDA kernels and do not support d=4.
> > >
> > > To enable a fair comparison at d=8, we aligned with AQLM's 2×8 configuration and added a kernel for 2×[256, 8] additive quantization. The throughput is nearly identical for d=4 and d=8, confirming that increasing the group size does not introduce noticeable overhead.
> > >
> > > In contrast to the highly specialized and fragmented kernel support of existing VQ methods, LC-QAT's affine dequantization is structurally simple and generalizes across group sizes with the same fused kernel, achieving the highest throughput among all VQ methods. We will include this analysis in the revision.

---

### Official Review · Reviewer_azh9 · 2026-03-07

**Soundness:** 3
**Presentation:** 3
**Significance:** 2
**Originality:** 2
**Overall Recommendation:** 4
**Confidence:** 3

**Summary:**

This paper introduces LC-QAT, a new QAT framework designed to compress LLMs into just 2-bit representations without serious information loss. To overcome difficulties typically associated with vector quantization in training process, the authors replace non-differentiable discrete codebook lookups with mathematical operations that allow for smooth, end-to-end learning using standard backpropagation. By ensuring a high-quality initial state before training begins, moreover, this approach significantly reduces the overall optimization difficulty. Consequently, the primary contribution of this work is its exceptional data efficiency, allowing models to achieve or surpass the performance of existing QAT methods using only a tiny fraction of the training data.

**Compliance With Llm Reviewing Policy:**

Affirmed.

**Final Justification:**

While my initial concerns regarding the method's novelty were justified—a point echoed by others—the authors successfully resolved these and other technical issues. Given the compelling proof of data efficiency, scalability, and the positive consensus among all reviewers, my main concerns are fully addressed. I maintain my positive assessment.

**Key Questions For Authors:**

Nothing.

**Limitations:**

Yes.

**Strengths And Weaknesses:**

**Strengths**
 - This reviewer thinks the most important strength of this paper seems the remarkable data efficiency achieved by the proposed method. The authors successfully demonstrate that LC-QAT can recover and even improve model performance using only a tiny fraction (0.1% to 10%) of the training data required by existing QAT methods. The comparison with BitNet 2B4T is particularly impressive and serves as a strong highlight of the paper. While BitNet requires a massive 4 trillion tokens to achieve its capabilities, LC-QAT achieves competitive performance using only 4 billion tokens.
 - The authors perform a commendable job of supporting their claims. The inclusion of loss landscape visualizations clearly illustrates why LC-QAT's initialization provides a more favorable starting point for optimization compared to SQ methods. Furthermore, the ablation studies effectively validate the necessity of the proposed components, specifically the Differentiable Gradient Estimator (DGE) and the Integer Weight Preprocessing, proving that they are crucial for preventing loss spikes and ensuring stable training.

**Weaknesses**
 - While the experiments on 1.7B, 3B, and 8B models are solid, the empirical validation would be significantly strengthened by testing on a more diverse range of model sizes, particularly larger models (e.g., 14B, 30B, or 70B+ parameters). Recognizing the challenging training set-up for larger model, data efficiency of this method could be helpful for advanced experiments. Demonstrating that this high data efficiency and performance retention scale to truly massive LLMs is crucial for a quantization paper targeting state-of-the-art deployment.
 - The evaluation of the foundation models (Tables 1 and 2) relies heavily on a specific subset of zero-shot commonsense reasoning tasks. While the paper includes diverse benchmarks like MMLU, GSM8K, and HumanEval for the instruction-tuned models in Table 3, it lacks these comprehensive benchmark scores for the base models. Including MMLU and other rigorous reasoning benchmarks in the broader foundation model comparisons would provide a much more complete picture of the models' preserved capabilities.

Note : About the novelty of this method, I would like to defer to the opinions of other reviewers with deeper historical expertise in this specific subfield to make a final judgment. I'm not sure about whether there have been similar methods among a tons of quantization papers or not.

---

> ### Author Rebuttal · Authors · 2026-03-31
>
> We thank the reviewer for the positive assessment and constructive suggestions. We address each concern below.
>
> **W1: Limited model scale diversity — lack of larger model experiments.**
>
> To address this concern, we conducted an additional experiment on a 14B-parameter model using approximately 58M tokens (constrained by time and computation). Two encouraging findings emerge: (1) LC-QAT's advantages over baselines hold consistently at this larger scale; (2) larger models recover more easily from quantization degradation, exhibiting smaller performance loss relative to their FP16 baseline — consistent with the intuition that larger models have more parameters to absorb quantization error.
>
> |        | wiki  | c4    | ARC-C | ARC-E | BoolQ | HellaSwag | PIQA  | WinoGrande | Avg.  | Per. |
> | ------ | ----- | ----- | ----- | ----- | ----- | --------- | ----- | ---------- | ----- | ---- |
> | FP16   | 8.64  | 13.81 | 60.15 | 82.83 | 89.30 | 78.84     | 79.87 | 72.85      | 77.31 | 1.00 |
> | LC-PTQ | 11.41 | 16.85 | 51.62 | 77.69 | 87.16 | 69.82     | 77.42 | 71.59      | 72.55 | 0.94 |
> | LC-QAT | 10.08 | 15.79 | 54.69 | 79.25 | 85.90 | 73.66     | 78.51 | 73.88      | 74.32 | 0.96 |
>
> Due to time constraints during the rebuttal period, we were only able to complete the 14B experiment. The consistent trend across 1.7B, 3B, 8B, and 14B models provides meaningful evidence of scalability. We will include larger model experiments in the revision.
>
> **W2: Limited benchmark coverage for foundation model evaluation.**
>
> We agree that broader benchmark coverage would provide a more complete picture, although our foundation model comparisons (Tables 1 and 2) follow the established zero-shot commonsense reasoning protocol used by prior works such as ParetoQ and PV-Tuning.
>
> To address this, we evaluated on more challenging reasoning benchmarks. Results show that LC-QAT consistently outperforms PV-Tuning on these harder tasks as well:
>
> |       | FP16  | LC-QAT | PV-Tuning |
> | ----- | ----- | ------ | --------- |
> | MMLU  | 55.26 | 45.92  | 44.70     |
> | Ceval | 58.25 | 36.03  | 34.70     |
> | CMMLU | 56.95 | 36.46  | 35.56     |
>
> This is consistent with our hypothesis that higher-quality VQ initialization leads to better preservation of learned representations, which matters most on tasks requiring deeper reasoning.
>
> **Regarding novelty:** To our knowledge, no prior work has achieved fully differentiable end-to-end training of vector-quantized LLMs. The fundamental barrier is that VQ's discrete codebook lookup is non-differentiable, which has confined all existing VQ methods to PTQ only. The only attempt at VQ-QAT (PV-Tuning) bypasses this via asynchronous coordinate descent, which cannot synchronously update all parameters and saturates quickly (Figure 4). Our linear-constrained parameterization removes this barrier by reducing codebook lookup to rounding/clamping + linear projection, making standard backpropagation applicable to VQ for the first time.

---

> > ### Author Rebuttal · Reviewer_azh9 · 2026-04-03
> >
> > My concerns have been adequately addressed.

---

> > > ### Author Response · Authors · 2026-04-05
> > >
> > > We thank the reviewer for confirming that the concerns have been fully resolved. We will incorporate all discussed improvements into the revision.

---

### Official Review · Reviewer_BbGr · 2026-03-11

**Soundness:** 3
**Presentation:** 3
**Significance:** 3
**Originality:** 3
**Overall Recommendation:** 4
**Confidence:** 3

**Summary:**

This paper introduces a method to perform vector-quantized QAT by using a parameterized codebook and differentiable gradient estimator.    The proposed method, LC-QAT, achieves strong performance on small open source models, although it is not clear if LC-QAT strictly outperforms other QAT methods due to different amounts of data being used for each method.

**Compliance With Llm Reviewing Policy:**

Affirmed.

**Final Justification:**

I think this method is interesting, but the empirical results could probably be a bit stronger, especially compared to a strong PTQ baseline like QTIP. This is especially true since QTIP uses LDLQ, and this QAT algorithm is initialized with LDLQ. I have personally run the QTIP and YAQA baselines before and know that they do better than reported here, but I will not hold this against the authors since they were probably unfamiliar with those baselines. I also think the authors could have done some more exploration with next token loss vs KL beyond the initial result they reported above. I think this paper is very borderline but I will raise my score to a 4.

**Key Questions For Authors:**

See above.

**Limitations:**

See above.

**Strengths And Weaknesses:**

Strengths:
1. The parameterized codebook construction allows for a "meaningful" gradient on the parameter and thus effective codebook index.
2. The DGE and integer preprocessing steps improve performance.
3. The method generally performs well on downstream evals (perplexity, zeroshot tasks)

Questions and Weaknesses.
1. In Tables 2 (and possibly also Table 2), the QAT methods use different amounts of data. It is not clear if LC-QAT is strictly better than ParetoQ since ParetoQ achieves lower perplexity but uses almost 10X more data. This paper would benefit from a data-scaling analysis for both methods.
2. Figure 5 suggests that the proposed "Integer Preprocessing" and DGE are doing the heavy lifting in final model performance. It is not clear how well QAT with a scalar quantizer and these methods performs and how much improvement actually comes from the parameterized VQ codebook.
3. Does LDLQ initialization actually help? I would like to see an ablation on this.
4. Does the QuIP# baseline use finetuning or not? A better baseline would be QTIP or QTIP + YAQA.
5. Are the QAT methods trained on the task loss (next token prediction) or the KL to the original model?
6. What is the effective shape of the codebook? If we use incoherence processing or pre-rotate the model such that the weights are approximately Gaussian, then the optimal codebook would also be Gaussian shaped. The current construction does not seem to admit this. Likewise, what prevents one from constructing a DGE for some generating function of the E8 lattice (used in QuIP# and NestQuant) or a trellis code (QTIP)?

---

> ### Author Rebuttal · Authors · 2026-03-31
>
> We thank the reviewer for the detailed feedback. We address each concern below.
>
> **W1: Data scaling analysis for LC-QAT vs ParetoQ.**
>
> We agree that comparing across different data budgets is important. While ParetoQ achieves lower perplexity with ~10× more data, LC-QAT already achieves higher average zero-shot QA accuracy on both LLaMA-3 models (8B: 73.42 vs 72.07; 3B: 68.33 vs 66.82).
> Due to computational constraints, we conducted a data-scaling analysis on Qwen-3-1.7B:
>
> | #Tokens | ARC-C | ARC-E | BoolQ | HellaSwag | PIQA  | WinoGrande | Avg.  |
> | ------- | ----- | ----- | ----- | --------- | ----- | ---------- | ----- |
> | 1B      | 39.51 | 64.35 | 70.43 | 55.01     | 71.65 | 57.54      | 59.75 |
> | 4B      | 41.72 | 68.18 | 69.85 | 58.23     | 72.69 | 57.93      | 61.43 |
> | 10B     | 42.41 | 68.90 | 69.02 | 59.60     | 73.45 | 61.33      | 62.45 |
>
> Performance continues to improve beyond 4B tokens without saturation. A symmetric analysis for ParetoQ is not feasible since its training data is not publicly available. Our primary claim is *data efficiency*: LC-QAT achieves competitive or superior downstream accuracy at a fraction of the data cost.
>
> **W2: Contribution of VQ parameterization vs. DGE and Integer Preprocessing.**
>
> Two points: (1) Integer Preprocessing is *specific to LC-QAT* and not applicable to SQ-QAT. In SQ, proxy weights are natural floating-point approximations needing no alignment. In LC-QAT, proxy weights must be initialized from discrete codebook indices — preprocessing is a necessary bridge for VQ-QAT, not a standalone improvement. (2) We applied DGE to a 2-bit SQ baseline and found no significant gain, consistent with Section 3.3: SQ initialization is far from any local minimum, so STE noise is not the bottleneck. This confirms LC-QAT's advantage stems from the VQ initialization.
>
> | task     | arc_challenge | arc_easy | boolq | hellaswag | piqa  | winogrande |
> | -------- | ------------- | -------- | ----- | --------- | ----- | ---------- |
> | GPTQ     | 26.79         | 26.47    | 44.71 | 26.55     | 52.01 | 47.75      |
> | GPTQ+DGE | 19.62         | 32.49    | 39.72 | 27.28     | 54.79 | 51.30      |
>
>
> **W3: Ablation on LDLQ initialization.**
>
> We replaced LDLQ initialization with random initialization. The result shows complete failure to converge, confirming that a high-quality PTQ starting point is essential.
>
> | Method  | ARC-C | ARC-E | BoolQ | HellaSwag | PIQA  | WinoGrande | Avg.  | Per. |
> | ------- | ----- | ----- | ----- | --------- | ----- | ---------- | ----- | ---- |
> | LC-PTQ  | 45.82 | 72.35 | 85.07 | 63.18     | 74.16 | 67.09      | 67.95 | 0.92 |
> | w/o LDLQ| 27.13 | 25.55 | 56.88 | 26.22     | 51.69 | 49.57      | 39.51 | 0.54 |
>
> **W4: QuIP# finetuning and comparison with QTIP baseline.**
>
> The QuIP# baseline in our experiments does not use finetuning; it is a pure PTQ result. We also added QTIP (Qwen-3-8B, 2-bit):
>
> | Method | PPL   | ARC-C | ARC-E | BoolQ | HellaSwag | PIQA  | WinoGrande | Avg.  | Per. |
> | ------ | ----- | ----- | ----- | ----- | --------- | ----- | ---------- | ----- | ---- |
> | QTIP   | 11.55 | 52.05 | 77.06 | 84.95 | 68.14     | 77.04 | 67.25      | 71.08 | 0.96 |
> | LC-PTQ | 14.82 | 45.82 | 72.35 | 85.07 | 63.18     | 74.16 | 67.09      | 67.95 | 0.92 |
> | LC-QAT | 10.23 | 53.75 | 78.82 | 82.29 | 71.37     | 76.99 | 69.85      | 72.18 | 0.97 |
>
> LC-QAT surpasses QTIP after training, and its decoding reduces to simple matrix multiplication, making it more practical. Regarding QTIP+YAQA: YAQA is a post-hoc fine-tuning method orthogonal to the quantization scheme and can equally be applied on top of LC-QAT.
>
> **W5: Training objective.**
>
> LC-QAT uses the standard next-token prediction loss. We will clarify this in the revision.
>
> **W6: Codebook shape and applicability to lattice/trellis codes.**
>
> *Codebook shape:* A is initialized as a scaled random orthogonal matrix (Eq. 4). By orthogonal transformation of uniform discrete inputs, codebook entries are approximately Gaussian-distributed — consistent with the near-Gaussian weight distribution from incoherence processing.
>
> *Why lattice codes (QuIP#, NestQuant) are unsuitable for end-to-end training:* Trainability requires dequantization to be a linear function of a regular integer grid. Lattice VQ violates this: (1) it selects a non-uniform subset of integer vectors for shaping, so indices cannot be recovered by rounding/clamping; (2) hierarchical constructions (E8P, nested lattices) cannot be collapsed into a single affine map, making differentiable training incompatible.
>
> *Why trellis codes (QTIP) are unsuitable:* Trellis codes require sequential Viterbi-style decoding. Integrating this into a training forward pass would incur substantial overhead, making it impractical for QAT. LC-QAT's linear decoding is fully parallelizable and computationally lightweight.
>
> The linear-constrained codebook balances representational quality and training tractability — a design space that lattice and trellis codes cannot occupy.

---

> > ### Author Rebuttal · Reviewer_BbGr · 2026-04-01
> >
> > Thanks for the response. A few questions:
> >
> > 1. Is the QTIP baseline using finetuning or not? If not, since LC-QAT is a QAT method, I think a fairer comparison would be to use the finetuned versions of QuIP#/QTIP. YAQA is also not a finetuning method. It is a different adaptive rounding Hessian.
> > 2. How does LC-QAT perform if you use the KL instead of next token loss?
> >
> > I think this paper is interesting and would be happy to increase my score if these questions are answered.

---

> > > ### Author Response · Authors · 2026-04-05
> > >
> > > Thank you for the follow-up questions and for the positive remarks about our work.
> > >
> > > **Q1: Finetuned QTIP/YAQA baselines.**
> > >
> > > We thank the reviewer for the correction — YAQA is indeed an adaptive rounding method, not a finetuning method. We apologize for the inaccuracy in our previous response.
> > >
> > > We have added comparisons with QTIP and YAQA, both with and without additional fine-tuning. Results below:
> > >
> > > *Qwen3-8B (2-bit):*
> > >
> > > | Method       | Wiki  | ARC-C | ARC-E | BoolQ | HellaSwag | PIQA  | WinoGrande | Avg.  | Per. |
> > > | ------------ | ----- | ----- | ----- | ----- | --------- | ----- | ---------- | ----- | ---- |
> > > | QTIP         | 11.55 | 52.05 | 77.06 | 84.95 | 68.14     | 77.04 | 67.25      | 71.08 | 0.96 |
> > > | QTIP+tuning  | 10.39 | 52.82 | 77.31 | 85.24 | 70.85     | 77.26 | 68.75      | 72.04 | 0.97 |
> > > | YAQA         | 11.51 | 52.39 | 78.28 | 82.57 | 68.62     | 76.61 | 68.35      | 71.14 | 0.96 |
> > > | YAQA+tuning  | 10.50 | 53.16 | 78.41 | 83.27 | 71.41     | 77.42 | 69.14      | 72.14 | 0.97 |
> > > | LC-PTQ       | 14.95 | 45.82 | 72.35 | 85.07 | 63.18     | 74.16 | 67.09      | 67.95 | 0.92 |
> > > | LC-QAT       | 10.23 | 53.75 | 78.82 | 82.29 | 71.37     | 76.99 | 69.85      | 72.18 | 0.97 |
> > >
> > > *Qwen3-1.7B (2-bit):*
> > >
> > > | Method       | Wiki  | ARC-C | ARC-E | BoolQ | HellaSwag | PIQA  | WinoGrande | Avg.  | Per. |
> > > | ------------ | ----- | ----- | ----- | ----- | --------- | ----- | ---------- | ----- | ---- |
> > > | QTIP         | 21.54 | 34.81 | 55.09 | 75.84 | 50.66     | 68.12 | 57.70      | 57.04 | 0.89 |
> > > | QTIP+tuning  | 18.21 | 36.09 | 59.13 | 76.39 | 53.36     | 69.64 | 58.96      | 58.93 | 0.92 |
> > > | YAQA         | 19.69 | 36.35 | 59.89 | 77.89 | 52.17     | 68.88 | 58.64      | 58.97 | 0.92 |
> > > | YAQA+tuning  | 17.14 | 37.03 | 61.99 | 78.41 | 54.83     | 70.24 | 59.27      | 60.30 | 0.94 |
> > > | LC-PTQ       | 29.11 | 31.91 | 53.87 | 67.46 | 44.99     | 61.80 | 54.38      | 52.40 | 0.82 |
> > > | LC-QAT       | 13.44 | 41.72 | 68.18 | 69.85 | 58.23     | 72.69 | 57.93      | 61.43 | 0.96 |
> > >
> > > On Qwen3-8B, finetuned QTIP and YAQA reach performance comparable to LC-QAT (all at 0.97 Per.), showing that at larger model scales, strong PTQ methods with fine-tuning are already competitive. On Qwen3-1.7B, where limited model capacity makes quantization error harder to absorb, LC-QAT (0.96) outperforms finetuned QTIP (0.92) and YAQA (0.94) by a clear margin. Moreover, as shown in our data-scaling analysis, LC-QAT's performance has not yet saturated, meaning it can continue to benefit from more training data. We will incorporate these comparisons into the revision.
> > >
> > > **Q2: KL divergence vs. next-token prediction loss.**
> > >
> > > We conducted an experiment using KL loss with 1B tokens on Qwen3-1.7B:
> > >
> > > | Loss | ARC-C | ARC-E | BoolQ | HellaSwag | PIQA  | WinoGrande | Avg.  |
> > > | ---- | ----- | ----- | ----- | --------- | ----- | ---------- | ----- |
> > > | KL   | 36.52 | 60.14 | 74.22 | 50.27     | 69.53 | 57.79      | 58.08 |
> > > | CE   | 35.58 | 62.79 | 68.34 | 51.92     | 70.89 | 57.85      | 57.90 |
> > >
> > > KL loss yields a slightly higher average accuracy (58.08 vs 57.90), while the two losses show different strengths across individual tasks, suggesting they may be complementary. We believe this is a promising direction and will discuss it in the revision.
> > >
> > > We hope these results fully address the remaining questions. We are happy to continue the discussion if needed.

---

### Decision · Program_Chairs · 2026-04-30

**Decision:**

Accept (regular)

**Comment:**

This paper proposes LC-QAT, a 2-bit weight-only quantization-aware training framework for LLMs. It introduces a linear-constrained codebook to enable a differentiable, end-to-end optimization without explicit codebook lookup forward pass, achieving high data efficiency. Experiments show that it matches or surpasses prior QAT methods using only a fraction of the training data.



The main strengths of this paper are:
- Interesting parameterized codebook effectively enables end-to-end differentiable training of VQ (Reviewer BbGr, LGNi), and clever forward pass reformulation (Reviewer Lu6C).
- Performance improvement and that on downstream evals (Reviewer BbGr).
- Remarkable data efficiency using only a tiny fraction of the training data compared to existing QAT methods (Reviewer azh9, LGNi, Lu6C).
- Solid ablations and visualizations support the necessity of the proposed components (DGE, Integer Preprocessing) (Reviewer azh9).


The main weaknesses are:
- Lack of a data-scaling analysis, the improvement of the parameterized VQ codebook, more ablations (like LDLQ initialization, group size, A/B sharing granularity, calibration dataset) and more details (like QuIP# baseline, QAT methods and codebook shape) (Reviewer BbGr, LGNi).
- Need for validation on larger models (>8B) and more comprehensive base model benchmarks (Reviewer azh9).
- Concerns about fair baselines, especially PV-Tuning's weaker initialization vs. LC-QAT's strong LC-PTQ start (Reviewer LGNi, Lu6C), and data amount discrepancies with ParetoQ (Reviewer BbGr).
- Insufficient discussion on hardware support and hardware-aware latency profiling. (Reviewer LGNi, Lu6C).


After the rebuttal, this paper received 3 Weak Accept and 1 Weak Reject. While most of the reviewers' concerns have been adequately addressed, the paper would still benefit from a clearer attribution of the performance gains, alongside more thorough ablation studies and explanations concerning the codebook construction, the strong PTQ baseline like QTIP，initialization process, and the QAT mechanism itself. The AC acknowledges the paper's data efficiency, and scalability. This paper is borderline, and the AC leans towards Accept. The authors are strongly encouraged to incorporate a clearer attribution of the performance gains as well as more ablation studies and limitations in the final version.